# The complex hybrid origins of the root knot nematodes revealed through comparative genomics

David H. Lunt[1], Sujai Kumar[2], Georgios Koutsovoulos[2] and Mark L. Blaxter[2,3]

[1] School of Biological, Biomedical and Environmental Sciences, University of Hull, Hull, UK
[2] Institute of Evolutionary Biology, University of Edinburgh, Edinburgh, UK
[3] The GenePool Genomics Facility, School of Biological Sciences, University of Edinburgh, Edinburgh, UK

Corresponding author
David H. Lunt,
dave.lunt@gmail.com

## ABSTRACT

Root knot nematodes (RKN) can infect most of the world's agricultural crop species and are among the most important of all plant pathogens. As yet however we have little understanding of their origins or the genomic basis of their extreme polyphagy. The most damaging pathogens reproduce by obligatory mitotic parthenogenesis and it has been suggested that these species originated from interspecific hybridizations between unknown parental taxa. We have sequenced the genome of the diploid meiotic parthenogen *Meloidogyne floridensis*, and use a comparative genomic approach to test the hypothesis that this species was involved in the hybrid origin of the tropical mitotic parthenogen *Meloidogyne incognita*. Phylogenomic analysis of gene families from *M. floridensis*, *M. incognita* and an outgroup species *Meloidogyne hapla* was carried out to trace the evolutionary history of these species' genomes, and we demonstrate that *M. floridensis* was one of the parental species in the hybrid origins of *M. incognita*. Analysis of the *M. floridensis* genome itself revealed many gene loci present in divergent copies, as they are in *M. incognita*, indicating that it too had a hybrid origin. The triploid *M. incognita* is shown to be a complex double-hybrid between *M. floridensis* and a third, unidentified, parent. The agriculturally important RKN have very complex origins involving the mixing of several parental genomes by hybridization and their extreme polyphagy and success in agricultural environments may be related to this hybridization, producing transgressive variation on which natural selection can act. It is now clear that studying RKN variation via individual marker loci may fail due to the species' convoluted origins, and multi-species population genomics is essential to understand the hybrid diversity and adaptive variation of this important species complex. This comparative genomic analysis provides a compelling example of the importance and complexity of hybridization in generating animal species diversity more generally.

## INTRODUCTION

Root-knot nematodes (RKN) belong to the genus *Meloidogyne,* contain approximately 100 described species, and are globally important crop pathogens (*Moens, Perry & Starr, 2009*). The most frequent, widespread, and damaging species (*M. incognita, M. arenaria,* and *M. javanica*) are tropical RKN that are highly polyphagous, infecting crop species producing the majority of the world's food supply, with the damage attributable to RKN ~5% of world agriculture (*Taylor & Sasser, 1978*; *Trudgill & Blok, 2001*; *Sasser & Carter, 1985*). The adaptive phenotypic diversity of these pathogens is also remarkable, with great variability observed both within and between species with respect to host range and isolate-specific vulnerability to control measures (*Trudgill & Blok, 2001*; *Castagnone-Sereno, 2006*). The tropical RKN typically reproduce by obligatory mitotic parthenogenesis and possess aneuploid genomes (*Triantaphyllou, 1982*; *Triantaphyllou, 1985*). These species have previously been suggested to be hybrid taxa, and phylogenetic analysis of nuclear loci supports this conclusion (*Dalmasso & Berge, 1983*; *Triantaphyllou, 1985*; *Hugall, Stanton & Moritz, 1999*; *Castagnone-Sereno, 2006*; *Lunt, 2008*).

Hybrid speciation has a long history of study in plants, with hybrid species formation having had a very significant influence on our understanding of species formation, diversity, and adaptation (*Arnold, 1997*; *Soltis & Soltis, 2009*). By contrast hybridization has been thought to be much less common in animals, though the utilization of multilocus genetics, and more recently genomics, has increased interest in the consequences of animal hybridization and several reviews suggest that it is much more common and important than previously thought (*Mallet, 2007*; *Mallet, 2005*; *Bullini, 1994*; *Nolte & Tautz, 2010*; *Schwenk, Brede & Streit, 2008*; *Seehausen, 2006*). Although there have been repeated suggestions that the tropical ("Group 1") RKN might have hybrid origins, the parental species involved have never been identified. The phylogenies in *Hugall, Stanton & Moritz (1999)* and *Lunt (2008)* indicate that these parents (as represented by divergent sequence clusters within the apomictic RKN) are more closely related to each other than either is to *M. hapla*, though neither had a parental species within their sampling schemes. *Meloidogyne floridensis* is a plant pathogenic root knot nematode that was originally characterized as *M. incognita*, but has since been described as a separate species on the basis of its morphology and a unique *esterase* isozyme pattern (*Jeyaprakash et al., 2006*; *Handoo et al., 2004*). Despite both nuclear rRNA and mtDNA sequences placing it within the phylogenetic diversity of the tropical mitotic parthenogen (apomict) species (*Tigano et al., 2005*; *Holterman et al., 2009*) (Fig. 1), *M. floridensis* is a meiotic parthenogen (automict) with the standard chromosome count of the meiotically reproducing RKN species ($n = 18$), has bivalent chromosomes, and an observable meiotic division (*Handoo et al., 2004*). *M. floridensis* appears to suppress the second meiotic division which is a known form of automictic reproduction called first-division restitution and a pathway by which parental heterozygosity can be maintained (*Bell, 1982*, p. 40). With the exception of *M. floridensis*, all of the Group 1 RKN (*De Ley et al., 2002*; *Holterman et al., 2009*) are apomicts, unable to reproduce by meiosis, lacking bivalents, and exhibiting extensive aneuploidy. This phylogenetic distribution of reproductive modes, with *M. floridensis*

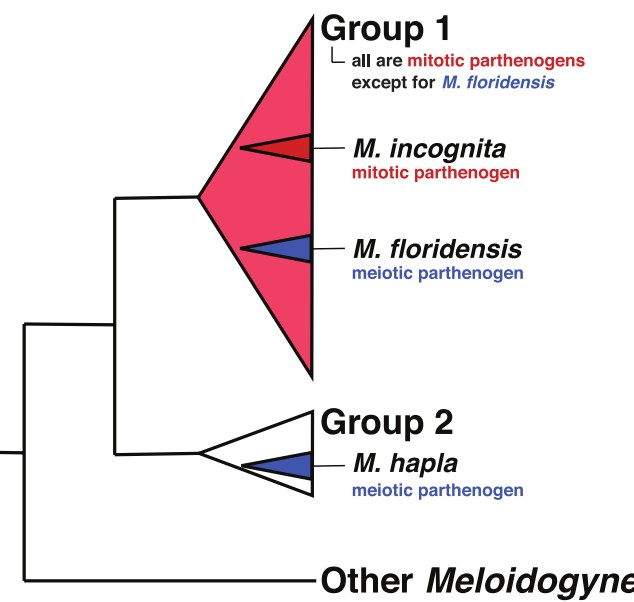

**Figure 1** **The relationships of tropical apomict *Meloidogyne*.** This image summarizes the relationships of the tropical apomict *Meloidogyne* root knot nematodes ("Group 1") to other *Meloidogyne*. *Meloidogyne floridensis* is a Group 1 species that can reproduce by meiotic parthenogenesis (blue colouration) while all other Group 1 species are obligate mitotic parthenogens (red colouration). *Meloidogyne hapla* is a meiotic parthenogenic species in Group 2. We have not used bifurcating trees to represent the relationships within the Group 1 and 2 species because of issues (highlighted in this paper) concerning possible hybrid origins of some taxa.

phylogenetically nested within the diversity of the apomict RKN (Fig. 1), is unanticipated as it implies the physiologically unlikely route of re-emergence of meiosis from within the obligate mitotic parthenogens. An alternative explanation for these observations is that the observed phylogenetic relationships have not arisen from a typical ancestor-descendent bifurcating process, but instead have been shaped by reticulate evolution and transfer of genes by interspecific hybridization with *M. floridensis* a parent of the tropical apomict species.

## The origins of *Meloidogyne incognita* genomic duplicates

The *M. incognita* genome revealed that many of the genes of this species are present as highly divergent copies (*Abad et al., 2008*), a situation that seems to apply to the other tropical apomicts too (*Lunt, 2008*), though the origin of these divergent copies is controversial. One possible way to account for the high divergence between alleles is that they have originated by a process of 'endoduplication' (Fig. 2A). Here we use endoduplication to refer to two distinct processes, although their genomic outcomes are similar. Firstly, the entire *M. incognita* genome might have doubled to become tetraploid. The homologous chromosomes may have then diverged, and the extant pattern of partial retention of duplicated loci could be the result of gene loss. This process would leave many areas of the genome possessing divergent copies. Second, an alternative mechanism possible in apomictic species such as *M. incognita*, is that former alleles

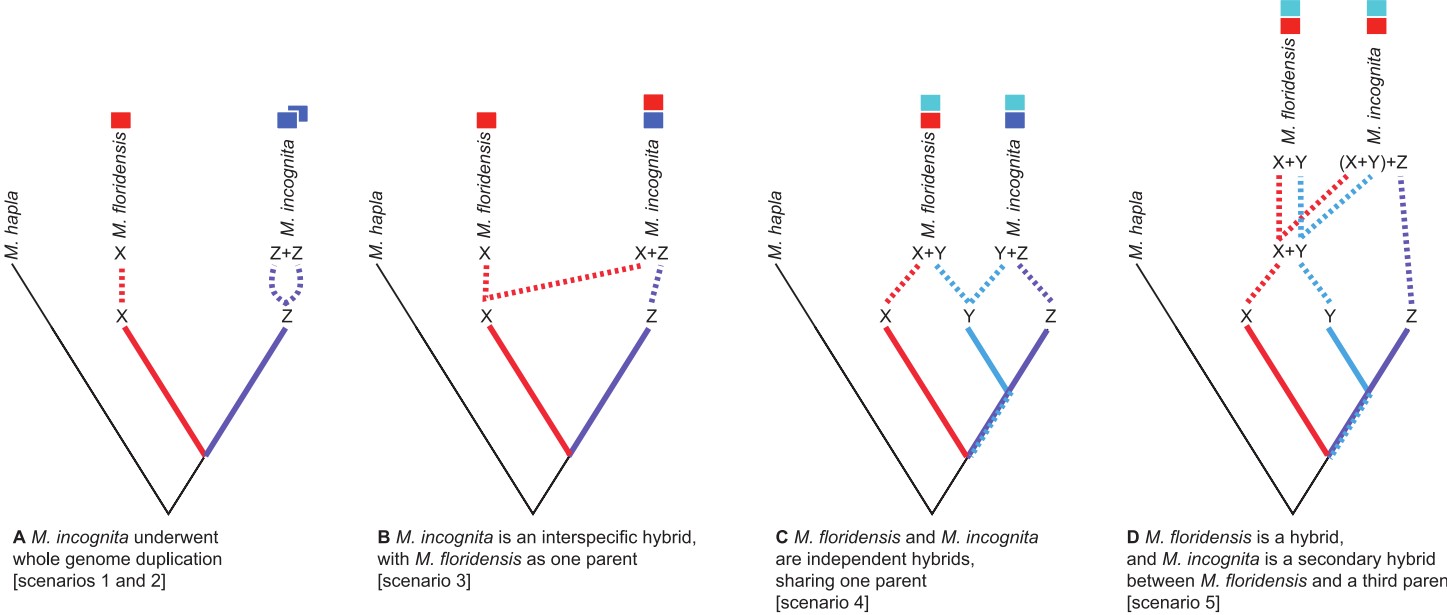

**A** *M. incognita* underwent whole genome duplication [scenarios 1 and 2]

**B** *M. incognita* is an interspecific hybrid, with *M. floridensis* as one parent [scenario 3]

**C** *M. floridensis* and *M. incognita* are independent hybrids, sharing one parent [scenario 4]

**D** *M. floridensis* is a hybrid, and *M. incognita* is a secondary hybrid between *M. floridensis* and a third parent [scenario 5]

**Figure 2** **Scenarios of the possible relationships between *Meloidogyne floridensis*, *Meloidogyne incognita* and *Meloidogyne hapla*, and the origins of duplicated gene copies.** *M. hapla* is a diploid species in a different sub-generic group to that of *M. incognita* and *M. floridensis*. Species "X", "Y" and "Z" are postulated ancestral parents that could have given rise to *M. incognita* and *M. floridensis*. (A) Scenarios 1 and 2: Here *M. floridensis* is a diploid sister species to *M. incognita* and possesses the "X" genome. Scenario 1 postulates reacquisition of apomixis in *M. floridensis* from an apomict ancestor, while Scenario 2 postulates that the apomicts repeatedly lost meiosis independently. Under both these scenarios, the presence of significant duplications in *M. incognita* suggests that it has undergone whole genome endoduplication. The duplicated genomes ("Z + Z") in *M. incognita* are diverging under Muller's ratchet. (B) Scenario 3: Ancestor "X" gave rise to the diploid species *M. floridensis*, and also interbred with "Z" to yield *M. incognita*, which thus carries two divergent copies of each gene ("X + Z"). In this model only *M. incognita*, not *M. floridensis*, is predicted to carry two homeologues of many genes. (C) Scenario 4: Both *M. floridensis* ("X + Y") and *M. incognita* ("Y + Z") are hybrid species, and share one parent ("Y"). In this model both *M. incognita* and *M. floridensis* are predicted to carry two homeologues of many genes. (D) Scenario 5: Both *M. floridensis* ("X + Y") and *M. incognita* ("X + Y + Z") are hybrid species, but *M. incognita* is a triploid hybrid between the hybrid *M. floridensis* ancestor ("X + Y") and another species ("Z"). In this model *M. incognita* is predicted to carry three, and *M. floridensis* is predicted to carry two, homeologues of many genes.

that are released from the homogenizing effects of recombination, can independently accumulate mutations over long periods of time resulting in highly divergent homologous loci ('alleles') within a diploid genome (*White, 1945*, pg. 283, *Judson & Normark, 1996*).

Another possible explanation for a genome containing divergent homologous copies of many genes is interspecific hybridization. One (homeologous) copy is inherited from each parental species and the divergence between them derives from the divergence between the hybridizing taxa. It is likely here that all genes would be present as divergent copies, although gene conversion and related processes could homogenize some copies. If it originated by this second mechanism the resulting *M. incognita* genome would be a mosaic with genomic regions derived from both its parents.

There are several ways in which *M. incognita* and *M. floridensis* might be related through hybridization. *M. floridensis* might be one of the two parental species which hybridized to form the tropical apomicts, including *M. incognita* (Fig. 2B). Alternatively, *M. floridensis* might be an independent hybrid that shares one parental taxon with *M. incognita*, and
thus represents a 'sibling' hybrid taxon (Fig. 2C). Finally, *M. floridensis* may itself be a hybrid, but still have played a role as a parent of *M. incognita* by a subsequent hybridization event (Fig. 2D). This last option predicts three gene copies in *M. incognita* and two in *M. floridensis*.

The nuclear gene phylogenies of *Lunt (2008)* indicate that the parental taxa of the apomict RKN were closely related and derived from within the cluster of Group 1 *Meloidogyne* species after the divergence of *M. enterolobii* (= *M. mayaguensis*). Since this closely matches the phylogenetic position of *M. floridensis*, which is known to reproduce via sexual recombination as the parental species also must have done, we set out to test by comparative genome sequencing and analysis if *M. floridensis* was one of the progenitors of the tropical apomicts.

## Reproductive mode and *Meloidogyne* evolutionary history

Given the unexpected distribution of meiosis across Group 1 *Meloidogyne* species described above (Fig. 1), there are several possible evolutionary pathways for the evolution of reproductive modes (Fig. 2): In scenario 1, *M. floridensis* has regained meiosis from an apomict state. Alternatively (scenario 2), the numerous apomict species could have lost meiosis many times independently. There are several additional scenarios involving hybrid origins. In scenario 3, the apomicts have hybrid origins with the automict *M. floridensis* as a putative parent, while in scenario 4 both *M. floridensis* and the apomicts have independent hybrid origins. In scenario 5, a hybrid *M. floridensis* is in turn parental to a complex hybrid apomict.

Scenario 1 is very unlikely. Meiosis is an exceptionally complex system to re-evolve once it has been lost (Dollo's law), and the only suggested example we are aware of in the literature is not supported by robust reanalysis (see *Goldberg & Igić, 2008*). In addition, the extant apomicts are highly aneuploid, making it necessary for *M. floridensis* to have re-evolved 18 homologous chromosome pairs, which again suggests that cytologically this route is highly unlikely. Scenario 2 is also not parsimonious, potentially implying very many independent major reproductive transitions. Since there are already genetic data indicating that the apomicts may have hybrid genomes (*Lunt, 2008*), we focused our analyses on the much more biologically plausible scenarios 3, 4 and 5 that propose hybridization drove the evolution of the apomictic RKN.

Scenario 3 restricts the hybrid taxa to the apomict Group 1 species, and places *M. floridensis* as one of the hybridizing parental species (Fig. 2B). This model makes predictions that, where divergent homeologous sequences are detected in the *M. incognita* genome, *M. floridensis* would possess two alleles closely related to one of these homeologues. The *M. floridensis* genome itself would also be substantially different from that of *M. incognita*, not possessing divergent homeologous blocks but rather displaying normal allelic variation, perhaps more similar to that observed in the *M. hapla* genome.

In scenarios 4 (Fig. 2C) and 5 (Fig. 2D) *M. floridensis* would also be a product of an interspecific hybridization, as are the apomicts. Both these scenarios predict that the *M. floridensis* genome will, like *M. incognita*, show substantial sequence divergence

between homeologues throughout its genome, although it may also possess some regions where one parental copy has been eliminated, and remaining diversity is simple allelism. In scenario 4, the parents of *M. incognita* need not be the same as those of the apomicts, although the phylogenetic position of *M. floridensis* implies that at least one of them may have been identical or very closely related. The different putative hybrid origins of *M. incognita* predict two (scenario 4, Fig. 2C) or three (scenario 5, Fig. 2D) homeologous copies, potentially modified by subsequent loss events.

Here, we generate a *de novo* assembled genome for *M. floridensis*, identify and analyse a large number of sets of homologous sequences in *M. floridensis*, *M. incognita* and *M. hapla*, and use both gene copy number distributions and gene phylogenies to test the predictions of the different scenarios outlined in Fig. 2.

## MATERIALS AND METHODS

### Nematode materials

DNA from female egg mass cultures of *Meloidogyne floridensis* isolate 5 was generously sourced and provided from culture by Dr. Tom Powers (University of Lincoln, Nebraska, USA) and Dr. Janete Brito (Florida Department of Agriculture and Consumer Services, Gainesville, USA).

### Sequencing and draft genome assembly

*Meloidogyne floridensis* DNA was prepared for sequencing using standard Illumina protocols by the GenePool Genomics Facility of the University of Edinburgh. A 260 bp insert library was sequenced using one lane of an Illumina HiSeq2000 (v2 reagents) with 101 base paired-end sequencing. 14.5 gigabases (Gb) of raw sequence data were adapter trimmed and quality filtered using perl and bash scripts to yield 70.2 M pairs totaling 13.2 Gb.

The genomic DNA sample derived from nematodes isolated from plant roots, and surrounded, therefore, by the bacterial communities of the rhizosphere. Egg masses of RKN are known to be associated with microbial taxa. To identify potential contaminants, we performed a preliminary assembly of all the trimmed reads ignoring pairing information. We then estimated read coverage of each assembled contig by mapping all reads back to the assembly, and annotated 10,000 randomly sampled contigs with the taxonomic order of their best megablast (BLAST+ version 2.2.25+ (*Zhang et al., 2000*)) match to the NCBI nt database (*Benson et al., 2011*). A taxon-annotated scatter plot of the GC% and coverage of each contig was used to visualize the contaminants present in the data (Fig. S1) (*Kumar & Blaxter, 2012*). Distinct GC%-coverage clusters in this plot were annotated with distinct taxonomic matches. A major cluster annotated as nematode was clearly dominant. Additional minor clusters were annotated as deriving from the bacterial orders Bacillales, Burkholderiales, Pseudomonadales and Rhizobiales. These all either had much lower coverage or much higher GC content than the nematode cluster. We conservatively removed contigs that matched the GC content and coverage of the identified contaminant blobs. To ensure optimal contamination removal, a second round

of megablast searches was performed and any contigs that matched Bacterial databases were removed. Only reads mapping to the remaining, putatively nematode contigs and their pairs were retained for the next step. The true insert size distribution of these reads was also estimated by mapping the pairs back to the preliminary assembly.

A stringent reassembly of the cleaned read set (11.1 Gb) was performed using reliable coverage information estimated from the preliminary assembly GC%-coverage plot. Velvet v1.1.04 (*Zerbino, 2010*; *Zerbino & Birney, 2008*) was used with a k-mer value of 55 and the parameters -exp_cov 45, -cov_cutoff 4.5, and -ins_length 260. Other parameters and assemblers were also tried but this assembly had the best contig length optimality scores (e.g., N50, the contig length at which 50% of the assembly is in contigs of that length or greater) and the highest CEGMA values (using CEGMA version 2.3, *Parra, Bradnam & Korf, 2007*). Redundant contigs likely to derive from independent assembly of allelic copies were removed using CD-HIT-EST (version 4.5.5, *Li & Godzik, 2006*) with -c 0.97 (removing all contigs that were more than 97% identical over their entire length to another, longer contig).

## Protein predictions and comparisons

A full annotation of the *M. floridensis* draft genome was not carried out, because no transcriptome data for the species was available. Instead, because we were interested in comparing coding sequences conserved with *M. hapla* and *M. incognita*, we used the protein2genome model in exonerate v2.2.0, (*Slater & Birney, 2005*) to align all *M. hapla* and *M. incognita* proteins, derived from the published genome sequences, to the *M. floridensis* draft genome. We extracted coding sequences (CDSs) that aligned to at least 50% of the length of the query protein sequences. If multiple *M. hapla* or *M. incognita* query protein sequences aligned to overlapping loci on the *M. floridensis* genome, only the longest locus was chosen as a putative *M. floridensis* CDS. The CDSs for all three species were trimmed after the first stop codon, and only sequences with a minimum of 50 amino acids were retained for further analysis.

To assess the level of self-identity among CDSs in each species, a BLASTn (version 2.2.25 +*Altschul et al., 1990*) search (with a sensitive E-value cutoff of 1e-5) was performed and the top scoring hit for each sequence to a CDS (other than itself) was selected if the length of the alignment was longer than 70% of the query sequence. The transcriipts of *M. incognita* were compared to the genomes of *M. floridensis* and *M. hapla* to identify levels of between species similarity using the same strategy.

## Clustering

We used Inparanoid (version 4.1, *Ostlund et al., 2010*) and QuickParanoid (*Kim & Park*) with default settings to assign proteins from the three *Meloidogyne* species to orthology groups. While assessing the level of duplication within the CDS sets (Fig. 3), we noted that several *M. incognita* CDS sequences were identical or nearly identical (>98% identity). These are most likely derived from allelic variants rather than gene duplications (which show a separate peak between 95 and 97% identity). To simplify the construction of orthologous gene clusters, we reduced these near identical sequences in each species using

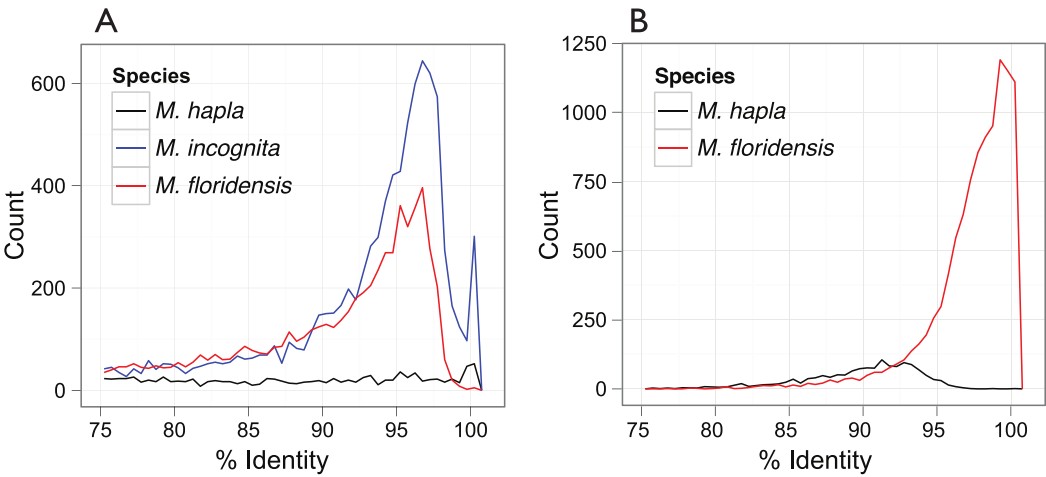

**Figure 3 Inter- and intra-genomic identification of duplicated protein-coding regions.** (A) Each coding sequence from each of the three target genomes (*M. hapla*, *M. incognita* and *M. floridensis*) was compared to the set of genes from the same species. The percent identity of the best matching (non-self) coding sequence was calculated, and is plotted as a frequency histogram. Both *M. incognita* and *M. floridensis* show evidence of the presence of many duplicates, while *M. hapla* does not. (B) The *M. incognita* gene predictions were compared to the *M. floridensis* genome and the *M. hapla* gene set. For each *M. incognita* gene, the similarity of the top matches in each genome was assessed. *M. incognita* has many genes that are highly similar to those of *M. floridensis* (similarity >98%). This contrasts with the matches to *M. hapla*, where the modal similarity is ~92%, and there is no peak of high-similarity matches.

CD-HIT-EST, removing any CDSs that were at least 98% identical across their whole length to another CDS.

## Phylogenetic analyses

For each InParanoid cluster, Clustal Omega v1.0.3, (*Sievers et al., 2011*) was first used to align the protein sequences. Tranalign (from the Emboss suite, v6.2.0, *Rice, Longden & Bleasby, 2000*) was then used along with the protein alignment as a guide to align the nucleotide CDS sequences. Finally, RAxML v7.2.8, (*Stamatakis, 2006*) was used to create maximum likelihood trees for each set of aligned CDS sequences in three steps: (i) finding the best ML tree by running the GTRGAMMA model for 10 runs using the command "raxmlHPC-PTHREADS-SSE3 -m GTRGAMMA -s $a -# 10 -n $a -T 2"; (ii) getting the bootstrap support values for this tree by running the same model until the autoMRE convergence criterion was satisfied employed the command "raxmlHPC-PTHREADS-SSE3 -m GTRGAMMA -s $a -# autoMRE -n $a.b -T 2 -b 12345"; (iii) using the bootstrap trees to draw bipartitions on the best ML tree used the command "raxmlHPC-PTHREADS-SSE3 -m GTRCAT -f b -t RAxML_bestTree. $a -z RAxML_bootstrap. $a.b -n $a.l -T 2 -o mh". Gene trees with a BP support of 70% or more were included in the analysis. The resulting trees were imported into the R Ape package v2.8, (*Paradis, Claude & Strimmer, 2004*) to count the number of trees with the same topology. Datafiles, treefiles, and scripts for processing the trees and other data can be obtained from FigShare 10.6084/m9.figshare.978784.

## RESULTS

### The genome of *Meloidogyne floridensis*

The *M. floridensis* genome was assembled using 11.1 Gb of cleaned data (see Table 1, Fig. S1) from 116 M reads (an estimated ∼100X coverage), using Illumina HiSeq2000 100 base paired-end sequencing of 250 bp fragments. The raw read data have been submitted to the Short Read Archive as accession ERP001338, the final assembly file is available at EMBL, and a blast database, CDS download, and other resources are available at http://brock.bio.ed.ac.uk/M_floridensis/ and http://nematodes.org/genomes/meloidogyne_floridensis.

### Intra-genomic comparisons reveal high numbers of duplicate genes in *M. incognita* and *M. floridensis*

Analysis of the distribution of within-genome CDS matches (Fig. 3A) identified an unexpected excess of apparent duplication in *M. floridensis*. While the CDS set of *M. hapla* had a relatively low rate of duplication, and no excess of duplicates of any particular divergence level, both *M. incognita* and *M. floridensis* had many more duplicates and a peak of divergence between duplicates at 95 to 97% identity. *M. incognita* showed an additional peak at ∼100% identity likely due to a failure to collapse allelic copies of some genes by the original authors (*Abad et al., 2008*). Because of the way we constructed our draft genome assembly, collapsing high-identity assembly fragments before analysis, *M. floridensis* lacked a near complete identity peak. To test for divergent copies of mtDNA within *M. floridensis* we searched the genomic contigs with a *M. floridensis* 16S mitochondrial rRNA gene from the international sequence databases (Genbank accession: AY635609.1) using blastn. Different regions of this query matched to two contigs comprising 1087 nucleotides in total and we observed a divergence between query and contig of 4/1087 nucleotides or 0.37%.

The very high frequency of intragenomic duplicate copies with a consistent divergence level strongly suggest that either *M. floridensis*, like *M. incognita*, is a hybrid species, with contributions from two distinct parental genomes, or that it has undergone a whole genome duplication. These distinct possibilities are addressed below. Comparing CDS between species we identified a high frequency of near-100% identity between *M. incognita* and its best match in the *M. floridensis* genome (Fig. 3B). This pattern was not evident when *M. incognita* was compared to *M. hapla*.

### Distinguishing sibling from parent–child species relationships

We identified several models that might explain the observed levels of within-genome divergent duplicates in *M. incognita* and *M. floridensis* (Fig. 3A). Expectations of relative numbers of (homeologous) gene copies per species, and the phylogenetic relationships of these homeologue sets differ and allow us to distinguish between the models. Thus for example under scenario 3 (Fig. 2B) we test to determine if *M. incognita* has two divergent homeologous gene copies, one of which is phylogenetically very closely related to the (collapsed) allelic copies in *M. floridensis*. We therefore clustered the CDS of the three species using InParanoid, after removing all CDS encoding peptides less than 50 amino acids in length.

**Table 1** Summary statistics describing genome assemblies of *Meloidogyne*.

| Species | *Meloidogyne hapla* | *Meloidogyne incognita* | *Meloidogyne floridensis* |
|---|---|---|---|
| Source | NCSU/WormBase WS227 | INRA/WormBase WS227 | 959 Nematode Genomes Project |
| Data URL | ftp://ftp.wormbase.org/pub/wormbase/species/m_hapla/ | ftp://ftp.wormbase.org/pub/wormbase/species/m_incognita/ | http://downloads.nematodegenomes.org |
| Citation | *Opperman et al. (2008)* | *Abad et al. (2008)* | This work |
| Maximum scaffold length | 360,446 | 154,116 | 40,762 |
| Number of scaffolds | 3,452 | 9,538 | 81,111 |
| Assembled size (bp) | 53,017,507 | 82,095,019 | 99,886,934 |
| Scaffold N50[*] (bp) | 37,608 | 12,786 | 3,516 |
| GC% | 27.4 | 31.4 | 29.7 |
| CEGMA[**] completeness Full/Partial | 92.74/94.35 | 75.00/77.82 | 60.08/72.18 |
| Predicted proteins (used for clustering[***]) | 13,072 (12,229) | 20,359 (17,999) | 15,327 (15,121) |

Notes.

[*] N50, weighted median contig length; the contig length at which 50% of the assembled genome is present in contigs of that or greater length.

[**] CEGMA, Core Eukaryotic Genes Mapping Approach (*Parra, Bradnam & Korf, 2007*).

[***] Predicted proteins used for clustering and inferring phylogenies after filtering for length >50 amino acids (see Methods).

**Table 2** Numbers of *Meloidogyne floridensis* and *Meloidoigyne incognita* members in homeologue gene sets that have one *Meloidogyne hapla* member.

| | 0 *M. incognita* members | 1 *M. incognita* member | 2 *M. incognita* members | 3 *M. incognita* members | >3 *M. incognita* members |
|---|---|---|---|---|---|
| 0 *M. floridensis* members | 0 | 907 | 327 | 44 | 17 |
| 1 *M. floridensis* member | 2196 | 2189 | 920 | 102 | 40 |
| 2 *M. floridensis* members | 226 | 257 | 156 | 36 | 21 |
| 3 *M. floridensis* members | 17 | 17 | 20 | 7 | 14 |
| >3 *M. floridensis* members | 8 | 11 | 6 | 4 | 21 |

We defined 11,587 clusters that contained CDS from more than one species, and 4,018 that had representatives from all three species (Fig. S2) These represent a number and proportion similar to comparisons between other nematode species with complete genomes (e.g., 2501 clusters were previously identified containing representatives from four nematode genomes *Mitreva et al., 2011*). As *M. hapla* is not expected to have undergone whole genome duplication, and we find no evidence of an excess of diverged duplicates in the *M. hapla* genome, we selected homologous gene sets where the ancestral gene was likely to have been single-copy by excluding clusters with more than one *M. hapla* member, and those lacking *M. hapla* members. We classified these clusters by the numbers of *M. incognita* and *M. floridensis* genes they contained (Table 2; Fig. 4). The trees generated and the scripts used to parse them into the categories represented in Fig. 4 are available through FigShare 10.6084/m9.figshare.978784.

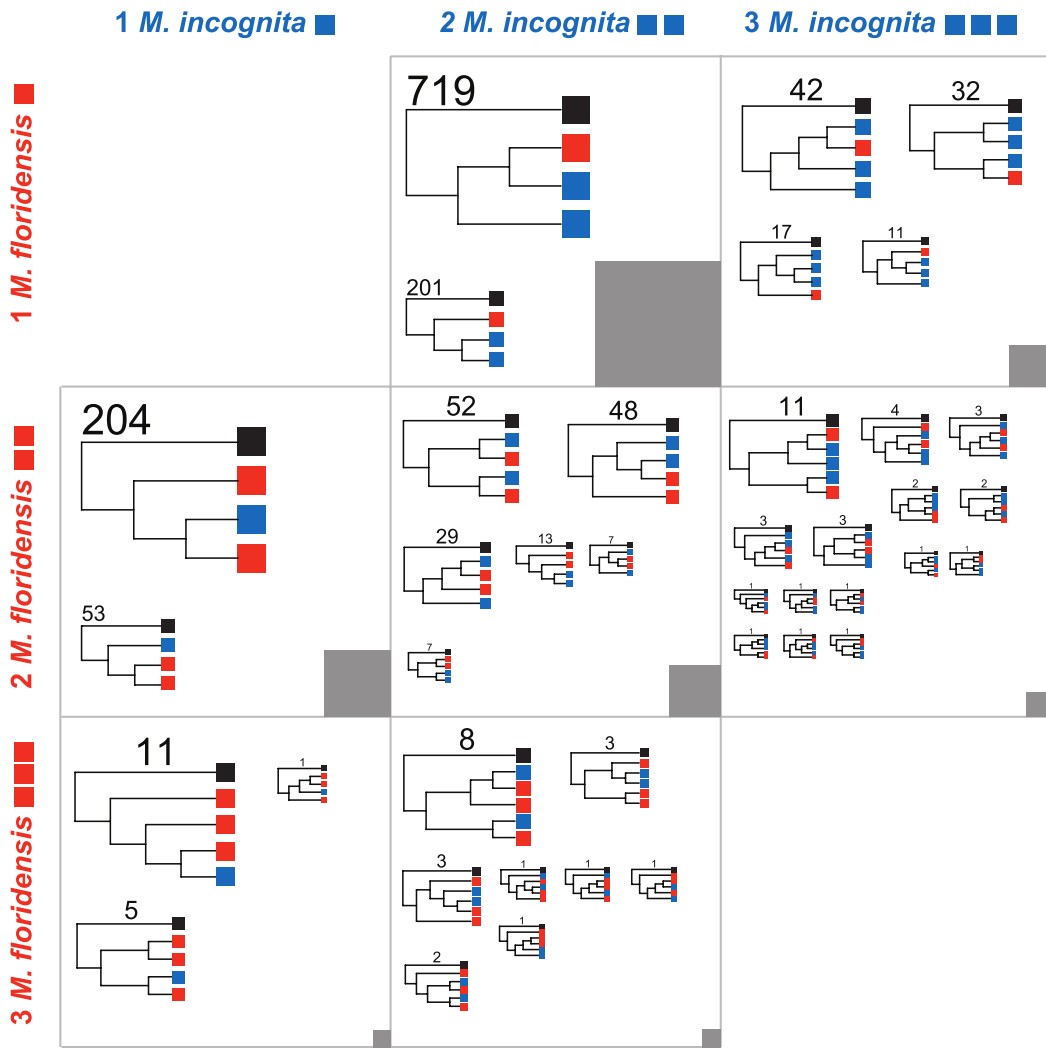

**Figure 4 Phylogenomic analyses of clustered gene sets.** For cluster sets represented in Table 2 that had representation of both *M. floridensis* and *M. incognita*, more than three members (i.e., where there was more than one possible topology), and fewer than five total members (i.e., where the number of possible topologies was still reasonably low and close to the number of clusters to be analyzed), we generated an estimate of the relationships between the sequences using RAxML. The resultant trees were bootstrapped, and rooted using the *M. hapla* representative. For each cluster set, the topologies were summarized by the different unique patterns possible. Within each figure cell, each cladogram in the figure is scaled by the number of clusters that returned that topology, with terminal nodes coloured by the origin of the sequences (black representing *M. hapla*, blue *M. incognita*, and red *M. floridensis*). The number of clusters congruent with each cladogram is given above the trees. The numbers of clusters contributing to each cell in the figure is represented by the grey box, which is scaled by the number of clusters summarized (e.g., the box in the central cell represents 902 trees, while the box in the bottom left cell represents 17 trees).

The process of idiosyncratic gene loss (or failure to capture a gene in the draft sequencing and assembly) is evident in the numbers of genes that have one *M. hapla* representative and no members from either *M. incognita* (column 1 of Table 2) or *M. floridensis* (row 1 of Table 2). Here it is striking that the clusters that contain only one *M. hapla* and one *M. floridensis* member (Mh1:Mf1:Mi0) outnumber by approximately

two to one clusters that have one *M. hapla* and one *M. incognita* member (Mh1:Mf0:Mi1). This suggests that the *M. floridensis* genome draft is a good substrate for these analyses (it contains homologues of many conserved genes apparently lost from, or missing in the draft assembly of, the *M. incognita* genome), and that the *M. incognita* draft is either incomplete or has experienced greater rates of gene loss.

The numbers of genes present in clusters that have two or more members, but lack one of *M. floridensis* or *M. incognita* (for example the 226 Mh1:Mf2:Mi0 clusters) reveal the potential extent of within-lineage duplication and divergence (and a component of stochastic loss of several homeologues in the missing species). There is no excess of these classes of cluster in *M. incognita*, arguing against a within-lineage, whole-genome duplication (i.e., against scenarios 1 or 2; Fig. 2A).

The striking feature of the membership of clusters (Table 2) is the number of cases where *M. incognita* has more cluster members than does *M. floridensis*. Thus there are 920 clusters in the class Mh1:Mf1:Mi2, but only 257 in the class Mh1:Mf2:Mi1, and 102 clusters in the class Mh1:Mf1:Mi3 compared to 17 in the class Mh1:Mf3:Mi1. This finding argues for the presence in *M. incognita* of at least one more genome copy than in *M. floridensis*, i.e., that *M. incognita* is likely to be a degenerate triploid hybrid (scenario 5, Fig. 2D). It is possible that some of the clusters in the Mh1:Mf1:Mi0 and Mh1:Mf0:Mi1 sets arise from *M. floridensis* and *M. incognita* being derived from different, divergent parents.

## Phylogenomic analysis of homologue relationships

A second set of predictions from the models in Fig. 2 concerns the phylogenetic relationships of the resulting sets of homologous gene sequences. Each model predicts a particular set of relationships between gene copies in each species. We therefore analyzed each informative set of clusters represented in Table 2 to identify which alternate topology was supported, assuming in each case that the single *M. hapla* representative was the outgroup. These phylogenomic results are summarized in Fig. 4. For each informative set of clusters, the majority topology supported scenario 5 (Fig. 2D), i.e., that *M. floridensis* is a hybrid, and was one of the parent species in a hybridization event that gave rise to a triploid *M. incognita*. Thus for the 902 Mh1:Mf1:Mi2 clusters, the topology in which one *M. incognita* CDS groups with the *M. floridensis* CDS to the exclusion of the other *M. incognita* sequence was favoured in 79% of the clusters, while in only 201 clusters (21%) the two *M. incognita* genes instead appeared to have arisen by duplication within *M. incognita*. In the Mh1:Mf2:Mi2 cluster set, one third of the clusters supported the topology where there were two independent sister relationships between *M. incognita* and *M. floridensis* genes. A further 48% of the trees were congruent with a triploid status for *M. incognita* where gene loss (or lack of prediction) had removed one *M. incognita* representative. The other classes of clusters could be interpreted in the same manner, and displayed trends that supported scenario 5.

## DISCUSSION

The genome structure and content of tropical *Meloidogyne* is revealed by our analyses to have had complex origins. It is likely that hybridization, ploidy change, and subsequent

aneuploidy have all played a role in the evolution of the diversity in this genus. The molecular evolutionary patterns revealed by comparative genomics however give us tools to conduct detailed analysis of these histories. This approach allows us to interpret the evolution of different reproductive strategies in terms of genome change, and better understand the evolution of these polyphagous pathogens.

### The *M. floridensis* genome reveals its hybrid origins

Our draft assembly of the genome of *M. floridensis* reveals a relatively typical nematode genome. The base haploid genome size for Meloidogyninae is likely to be ∼50 Mb. Both the sequenced genome of *M. hapla* (*Opperman et al., 2008*), and independent measurement of its genome size from densitometry (*Pableo & Triantaphyllou, 1989*), yield estimates of 50–54 Mb. The sequenced genome estimate is unlikely to be inflated through issues of uncollapsed haploid contigs, as *M. hapla* is expected to have reduced heterozygosity through its automictic reproductive mode (*Liu, Thomas & Williamson, 2007*), and the sequenced strain was inbred (*Opperman et al., 2008*). Hybrid taxa, containing homeologous chromosomes from more than one parental lineage, would be expected to have genome assembly sizes that are the sum of the parental genomes, albeit modified by idiosyncratic post-hybridization gene loss and repeat copy change. Thus the ∼100 Mb genome size estimated for *M. floridensis* is in keeping with a base Meloidogyninae genome of ∼50 Mb, with homeologous sequences assembled independently. The divergence between inferred homeologous genes in our genome (∼4–8%) would preclude coassembly of homeologous coding sequences, and the higher divergence found in intergenic and intronic sequences would make them even less likely to be coassembled. The published *M. incognita genome* is 86 Mb, but ongoing revision of the assembly suggests a true value of ∼130 Mb (E Danchin, pers. comm., 2014), as might be expected for a hypo-triploid species.

The *M. floridensis* genome assembly is less contiguous than those of *M. hapla* and *M. incognita* (reflected in the lower contiguity and content of conserved eukaryotic genes). Such fragmentation is a known limitation of using a single small-insert paired-end library, and refinement of the assembly using larger-insert mate pair, or long single molecule reads, would undoubtedly improve the biological completeness of the product. Our primary aim however was not to produce a highly contiguous assembly, but rather to identify protein-coding sequences (CDS) for use in comparative genomic analyses. Despite the fragmentation we were able to identify over 15,000 CDS segments to address the possible hybrid status of *M. floridensis* and *M. incognita*, making it more than sufficient for this study.

We note that both the *M. incognita* and the *M. floridensis* genomes have low scores (60–75%) when assessed by the Core Eukaryotic Genes Mapping Approach (CEGMA), compared to the 94% scored by the *M. hapla* assembly (and assemblies of other nematode genomes). However, the published *M. incognita* genome, while having much better assembly statistics (only 9,538 scaffolds, and a contiguity ∼4 times that achieved for *M. floridensis*), has similarly poor scores in CEGMA analysis. Whether this is a reflection

of shared divergent biology, or, as we suspect, a poor fragmented assembly, will require additional sequencing data, reassembly and reassessment.

The phylogenetic position of the automictic *M. floridensis* suggest that this species, or an immediate ancestor, was parental to the tropical apomicts, i.e., being one partner in the hybrid origins of the group (scenarios 3 and 5, Figs. 2B and 2D). It is also possible however that *M. floridensis* is not directly parental to the apomicts, but rather a hybrid sibling, also arising by interspecific hybridization (scenario 4, Fig. 2C). In this case one parent of *M. floridensis* is very likely to also have been involved in the hybrid origins of *M. incognita* as very many loci were found to be nearly identical between *M. incognita* and *M. floridensis* (Fig. 3B). In order to distinguish between scenario 3 (diploid parent), scenario 4 (hybrid sibling) and scenario 5 (hybrid parent) we examined the sequence diversity within each species' genome.

### Intra-genomic divergence of coding loci

Information concerning the hybrid status of *M. floridensis* can be gained from comparing the pattern of gene duplication within its genome to that of other RKN species, since *Meloidogyne incognita* has been suggested previously to have hybrid origins (*Dalmasso & Berge, 1983*; *Triantaphyllou, 1985*; *Hugall, Stanton & Moritz, 1999*; *Castagnone-Sereno, 2006*; *Lunt, 2008*) whereas *M. hapla* never has. An interspecific hybrid would be expected to have an excess of divergent intra-genomic duplicates compared to a non-hybrid, due to its homeologous chromosome pairs. The genome of *M. hapla*, a closely related species without a hybrid origin, represents the normal intra-genomic duplication pattern without homeologous chromosomes. In *M. hapla* there was a very much lower number of divergent duplicates compared to the other species, and these had a wide range of divergences rather than a frequency peak at any divergence value. While there was a slight excess of duplicates with high identity in *M. hapla*, the distribution overall is consistent with an ongoing rare process of stochastic duplication followed by gradual divergence (Fig. 3A).

In contrast to the pattern observed in *M. hapla*, the intra-genomic comparisons of both *M. incognita* and *M. floridensis* revealed many more divergent duplicated CDS (Fig. 3A). We observed a peak of high-identity duplicates in *M. incognita* that was absent in *M. floridensis*. This is most likely because we stringently collapsed high identity segments (as putative allelic copies) during assembly of *M. floridensis* whereas the *M. incognita* genome assembly may still contain some of these alleles. Most striking however was the presence in both species of a frequency peak of more diverged duplicates showing ∼96% identity. Such duplicates have been described in *M. incognita* (*Abad et al., 2008*; *Lunt, 2008*) although the scale of these diverged loci and their presence in *M. floridensis* has not been reported previously.

If the *M. floridensis* divergent copies were the product of a mixed sample, and thus represented polymorphism rather than homeologs, we would predict that mitochondrial DNA would also display this pattern. Our BLAST search however did not find any divergent contigs matching to our mtDNA genbank query. The query sequence differed by only 0.37% and this divergence between the genbank sequence and our strain is typical of intraspecific polymorphism levels in other nematodes but is approximately ten-fold less

than we observe between the nuclear divergent copies in the *M. floridensis* genome. We therefore do not consider that, even if our starting material had been contaminated with a second *M. floridensis* strain, such intraspecific polymorphism could account for our much more diverged genomic copies.

Ongoing individual gene duplication events—which we propose has generated the *M. hapla* distribution—could not have produced these patterns. Instead, the distributions are congruent with either a single major past event of genome duplication followed by divergence, or else hybridization to bring together pre-diverged homeologous chromosome copies that had been evolving independently since the last common ancestor of the parental species. On top of these processes differences in the rates of evolution of individual loci has resulted in variation in observed identity in the extant genomes, producing a distribution around a single peak of divergence. While these two alternative scenarios (endoduplication and homeologous chromosomes) cannot be distinguished on the basis of duplicate divergence data alone, the analysis does suggest that the genome content of both *M. floridensis* and *M. incognita* have been shaped in very similar ways by major duplication or divergence events.

### Integrating phylogenomic analyses

To distinguish between endoduplication and hybrid origins of these CDS divergences, we examined the phylogenetic histories of sets of homologous loci from the three *Meloidogyne* genomes. By selecting CDS clusters with only a single member from the *M. hapla* genome we have likely restricted our analyses to loci that were single copy in the last common ancestor of the three species, and thus do not show the complexities of turnover in large multigene families.

We compared support on a gene-by-gene basis for tree topologies that would support or refute the hybrid *versus* endoduplication scenarios (Fig. 2, Table 2, Fig. 4). Using this approach we could robustly exclude scenario 1, endoduplication of the *M. incognita* genome, as a source of duplicate CDS since we frequently observed that these *M. incognita* sequences were not monophyletic with respect to *M. floridensis*. If *M. incognita* had duplicated its own genome we would expect these duplicate CDS to share a recent origin and be each other's closest relatives. We could similarly exclude scenarios 2 and 3, since intra-genomic comparisons of CDS in the *M. floridensis* genome revealed that it also possesses divergent duplicates, and phylogenetic analyses indicated that these, just like the *M. incognita* sequences, are not monophyletic by species.

Thus we suggest that the most parsimonious explanation of the duplicate divergence and phylogenetic data is that both *M. floridensis* and *M. incognita* are hybrid species, and the duplicate CDS are homeologues rather than within-species paralogues. We can distinguish between scenario 4 (independent hybrid origins: the two species are step-sisters) and scenario 5 (*M. floridensis* represents one of the parents of a triploid hybrid *M. incognita*) by phylogenetic analyses of the clustered CDS. We observed an excess of clusters where there were more *M. incognita* members than there were *M. floridensis* members, as would be expected from a triploid species, whether or not it was now losing duplicated genes stochastically. In these clusters, the extra *M. incognita* CDS was

less likely to be sister to one of the other *M. incognita* CDS than it was to be a sister to a *M. incognita–M. floridensis* pair. Based on these data we suggest that the triplicate loci in *M. incognita* are the three homeologues that have resulted from a hybridization event between the hybrid *M. floridensis* and an unidentified second, likely non-hybrid, parent (scenario 5, Fig. 2D). For clusters containing two *M. floridensis* homeologues and two *M. incognita* homeologues, the topology supporting shared hybrid ancestry was again more frequently recovered than topologies supporting independent hybridization events.

*Handoo et al. (2004)* described the meiosis of *M. floridensis* as lacking a second maturation division and being 'intermediate' between meiotic and mitotic forms of reproduction. The division observed by *Handoo et al. (2004)* is in fact likely a long-known form of purely meiotic automixis called "first division restitution". *Bell (1982, p. 40)*, in his classic review of the evolution of mating systems, describes one of the three primary types of automixis as involving the suppression of the second meiotic division, exactly as described by *Handoo et al. (2004)* for *M. floridensis*. The maintenance of both homeologs in the *M. floridensis* genome through meiotic divisions, as we report here, may seem more challenging than in *M. incognita*, which reproduces only by mitosis. Automixis that maintains the parental heterozygosity is however well described in other animals and we assume a very similar mechanism occurs in *M. floridensis* (*Bell, 1982*, p. 40; *Smith, 1978*, p. 44; *Hood & Antonovics, 2004* and refs therein).

### Hybrid speciation and adaptive novelty

Animal hybrids have been characterized as rare, unfit, and adversely affected by both competition and gene flow from their parents (*Mayr, 1963*; *Barton, 2001*). There is now an increasing awareness in the literature however of animal hybridization as both a speciation mechanism and a route to the generation of novel phenotypic diversity on which natural selection may act (*Bullini, 1994*; *Arnold, 1997*; *Mavarez & Linares, 2008*; *Soltis & Soltis, 2009*; *Abbott et al., 2013*). There are a growing number of cases in which animal species have a hybrid origin, i.e.: it is known that all vertebrate constitutive parthenogens, and gynogenetic species have hybrid origins (*Avise, 2008*); the Italian sparrow (*Passer italiae*) has been shown to be a nascent hybrid species (*Hermansen et al., 2011*); hybridization between two species of *Rhagoletis* tephritid fruitflies has led to expansion into a novel ecological niche (host plant) in the hybrid, and also reproductive isolation from both parents since mating is confined to the host plant (*Schwarz et al., 2005*). The genetic basis of hybridization in generating adaptive diversity has been revealed in a number of studies: the *Heliconius melpomene* genome demonstrates that hybridization and introgression has been important for the adaptive radiation of these butterflies, by sharing protective colour-pattern genes among co-mimics (*Heliconius Genome Consortium, 2012*); the Northern European freshwater 'invasive sculpin' fish are hybrids between two geographically isolated *Cottus* species and they have colonized a novel niche consisting of the extensively human-altered lower reaches of the rivers Rhine and Scheldt (*Czypionka et al., 2012*). The cichlid adaptive radiation in Lake Malawi involves the evolution of more than 400 species, over a period of only 4.6 million years (*Genner et al., 2007*), which have colonized and adapted to many diverse lacustrine habitats. Recent genetic studies indicate

that this radiation, and cichlid diversification in general, has been strongly influenced by interspecific hybridization (*Joyce et al., 2011*; *Schwarzer et al., 2012*; *Loh et al., 2012*; *Genner & Turner, 2012*).

It has been suggested that hybrid animal taxa are most likely to succeed where new habitats open up, and such events may have played a significant role in several classic examples of adaptive radiation (*Seehausen, 2006*; *Abbott et al., 2013*; *Seehausen, 2013*; *Kearney, 2005*). The tropical RKN are exceptionally successful globally-distributed pathogens of diverse agricultural crops (*Moens, Perry & Starr, 2009*; *Trudgill & Blok, 2001*). These species have colonized a novel habitat, show extensive functional diversity, and have adapted to crop host-plants in the very brief evolutionary timeframe that agriculture has existed (a few thousand years). This is a situation similar to other animal adaptive radiations where hybridization may also have played a significant role (*Seehausen, 2006*; *Seehausen, 2013*; *Abbott et al., 2013*).

Although the adaptive consequences of hybridization are being increasingly recognized as important for biodiversity, ecology and evolution, the origin of novel traits, colonization of new ecological niches, and adaptive evolution can lead to serious problems if the organisms concerned are pathogens of humans, livestock, or crops (*Bisharat et al., 2005*; *Brasier, 2001*; *Stukenbrock et al., 2012*; *Inderbitzin et al., 2011*; *Goss et al., 2011*). It is particularly important therefore to understand the genetic basis of adaptive diversification in relation to existing or emerging pathogens.

The tropical apomictic RKN, exemplified by *M. incognita*, *M. arenaria* and *M. javanica*, possess host ranges that may include practically all agriculturally important species overlapping their distribution, causing *M. incognita* to be described as the "single most damaging crop pathogen in the world" (*Trudgill & Blok, 2001*). Such extreme polyphagy is not typically encountered in *Meloidogyne* species outside of the radiation of tropical apomicts, although some do exploit multiple hosts. The origins and mechanisms of this greatly expanded host range are not only interesting from an evolutionary genomics perspective but also important to our understanding of the mode of action of these globally important crop pathogens. The demonstration of the hybrid origins of *M. incognita* and *M. floridensis*, and by implication *M. javanica* and *M. arenaria* also, suggests transgressive segregation of adaptive variation might have played an important role in determining host range. Transgressive segregation is when the absolute values of traits in some hybrids exceed the trait variation shown by either parental lineage. Such transgressive phenotypes are common in hybrid offspring in both animals and plants, and particularly so where the parents derive from inbred but divergent lineages (*Rieseberg, Archer & Wayne, 1999*). Transgressive phenotypes have played a significant role in plant breeding, where crossing of inbred parental lineages can lead to extreme offspring variation onto which artificial selection is imposed, and similar processes are likely to act on hybrid swarms resulting from natural selection acting on inter-species crosses in the wild (*Rieseberg, Archer & Wayne, 1999*; *Stelkens & Seehausen, 2009*; *Genner & Turner, 2012*). We do not yet know whether transgressive phenotypes in hybrid apomict RKN have been shaped by natural selection, but given our increasing awareness of its importance in adaptive radiations, and

the frequency with which hybrid plant pathogens are detected in other systems (*Stuken-brock et al., 2012*; *Stukenbrock & McDonald, 2008*; *Inderbitzin et al., 2011*; *Brasier, 2001*), it may be an important direction for future research allowing us to detect likely pathogens at early stages.

Although we have not yet identified the parental taxa of *M. floridensis*, or the second parent of the tropical apomict RKN, it is likely that they were facultatively sexual meiotic parthenogens, as this is the most common reproductive mode within *Meloidogyne* (*Triantaphyllou, 1982*; *Triantaphyllou, 1985*; *Chitwood & Perry, 2009*). This breeding system can fuse the products of a single meiotic division in order to regain diploidy, making these taxa more similar to the inbred lineages of plants highlighted as frequent sources of transgressive segregation and extreme phenotypes (*Rieseberg, Whitton & Gardner, 1999*) than to the typical (amphimictic) species of hybridizing animals. If this "polyphagy as transgressive segregation" hypothesis were correct then we would predict that the parents of the polyphagous RKN would most likely be automicts with considerably smaller host ranges.

## Hybridization and molecular genetic approaches to *Meloidogyne* diversity

Molecular approaches to understanding the diversity of apomictic RKN have a long history and include studies of isozymes, mitochondrial DNA (mtDNA), ribosomal internal transcribed spacer (ITS), ribosomal RNA genes (rDNA), random amplified polymorphic DNA markers (RAPDs), amplified fragment length polymorphisms (AFLPs), and other marker systems (see *Blok & Powers, 2009* for a review). However, if some *Meloidogyne* species are in fact hybrids, this presents particular problems for the standard molecular approaches used to characterize diversity. These typically assume that species or isolates have diverged following a bifurcating, tree-like, evolutionary pathway. Hybridization violates this assumption and produces more complex evolutionary histories that can either be misrepresented by single locus markers, or else produce intermediate or equivocal signal from multi-locus approaches. For example, a major reason that mtDNA and rDNA sequencing have been useful in evolutionary ecology is that they are effectively haploid, and hybrid taxa, which often retain just one of their parental species' genotypes at these loci, present particular problems for these approaches (*Seehausen, 2006*; *Hailer et al., 2012*; *Meyer et al., 2012*). While carefully benchmarked marker approaches may still have utility in diagnostics, they will not be able to accurately reflect the complex evolutionary pathway of hybrid *Meloidogyne* species where different loci are likely to have experienced very different histories. Incongruence between markers is therefore to be expected as a true reflection of history, rather than due to a lack of analytical power. We are currently in the early stages of *Meloidogyne* comparative genomics and current estimates of the complex phylogenetic relationships between hybrid taxa will need to be constantly refined as more species are added.

Genomic approaches to the RKN system hold many advantages, including documenting the genomic changes associated with host-specialization, extreme polyphagy, and

interaction with plant defense systems. An interesting and important question now is whether the main apomictic RKN species have a single origin, with species divergence perhaps related to aneuploidy, or are instead the result of repeated hybridizations of the same or similar parental lineages. Different patterns of origin may determine the extent to which control strategies may be broadly or only locally applicable. Finally, if transgressive segregation is a cause of extreme and unique diversity, including polyphagy and novel resistance breaking isolates, then monitoring of new hybrid lineages may be an agricultural necessity. We are now close to the time where RKN isolates can be characterized not only with a trivial name (e.g., *M. incognita* race X) but instead a detailed list of genome wide variants and their known association with the environment, response to nematicides, and virulence against a range of plant host species and genotypes—an approach that will surely be extremely valuable in optimizing agricultural success. We caution therefore that although traditional genetic approaches may be valuable for rapid diagnostics, population genomics must be embraced in order to really advance our understanding of these important pathogens and maximize our ability to successfully intervene.

## CONCLUSIONS

Here we have used whole genome sequencing and evolutionary comparative genomics to demonstrate the complex hybrid origins of key Root Knot Nematode species. Understanding the evolutionary history of *Meloidogyne* species is a priority since only by this route can the evolution of pathogenicity and resistance, the emergence of new pathogenic strains, horizontal transfer of genes, and geographic spread of one of the world's most important crop pathogens be properly understood. The importance of animal hybridization to speciation and adaptation is being increasingly recognized, driven by new insights from genome sequencing. *Meloidogyne incognita* is shown to be an unusual double-hybrid, suggesting that hybridization may be a common and complex process in the history of this group. The *Meloidogyne* system, with its very recent expansion to fill numerous agricultural ecological niches, shows interesting parallels to natural adaptive radiations that may also have been greatly influenced by hybridization. Further work elucidating whether hybridization contributes adaptively to polyphagy will be important not just in the context of root knot nematodes, but also in determining the interplay of evolutionary forces generating organismal adaptive divergence more generally.

## ACKNOWLEDGEMENTS

We thank Tom Powers and Janete Brito for sourcing and supplying *M. floridensis* materials, Etienne Danchin for access to *M. incognita* genome data and Marian Thomson and members of the GenePool Genomics Facility for sequencing support. We thank Africa Gómez, Amir Szitenberg, Steve Moss, Richard Ennos and Karim Gharbi for comments on the manuscript and the project.

### Funding

SK was supported by an overseas Research Studentship award of the School of Biological Sciences, University of Edinburgh, and GK by a BBSRC Research Studentship and an ORS award. The GenePool has core support from the NERC (award R8/H10/56) and MRC (G0900740). DL and MB are supported in part by NERC award NE/J011355/1. The funders had no role in study design, data collection and analysis, decision to publish, or preparation of the manuscript.

### Grant Disclosures

The following grant information was disclosed by the authors:
Research Studentship award of the School of Biological Sciences, University of Edinburgh.
BBSRC Research Studentship.
ORS award.
NERC: R8/H10/56.
MRC: G0900740.
NERC: NE/J011355/1.

### Competing Interests

The authors declare there are no competing interests.

### Author Contributions

- David H. Lunt and Mark L. Blaxter conceived and designed the experiments, analyzed the data, contributed reagents/materials/analysis tools, wrote the paper, prepared figures and/or tables, reviewed drafts of the paper.
- Sujai Kumar and Georgios Koutsovoulos performed the experiments, analyzed the data, prepared figures and/or tables, reviewed drafts of the paper.

### DNA Deposition

The following information was supplied regarding the deposition of DNA sequences:
NCBI Short Read Archive accession ERP001338.

### Data Deposition

The following information was supplied regarding the deposition of related data:
EMBL: PRJEB6016, PRJEB295.

### Supplemental information

Supplemental information for this article can be found online at http://dx.doi.org/10.7717/peerj.356.

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
