# Peer review of "The complex hybrid origins of the root knot nematodes revealed through comparative genomics"

_PeerJ, doi:10.7717/peerj.356_

## Round 0.1 · original submission · Major Revisions

Both reviewers showed enthusiasm for your study and provided a wealth of constructive remarks. Please address them in a revised version of your manuscript. In particular, some of your claims might need to be hedged or toned down.

·

Basic reporting

This paper utilizes a comparative genomics approach to address a compelling question regarding an important group of plant pathogens. Interest in understanding how this asexually reproducing group of pathogens evolved and how it has become so successful and is able to adapt to new hosts comes from both agriculturalists searching for new management strategies following increasing restrictions on toxic pesticides and from evolutionary biologists attempting to understand mechanisms for genetic change in asexual species. The whole genome comparisons improve previous attempts to assess how these asexual species arose based on specific sequence comparison or polymorphism analyses. The paper adds the sequence of the root knot nematode Meloidogyne floridensis to the already available information on the genomes of the close relative M. incognita and the more distantly related species M. hapla. M. floridensis is a good choice as it appears to have a different chromosome composition and reproductive mechanism than does M. incognita.

However, some aspects of the background information presented in this paper are misleading and should be clarified. Most importantly, the authors refer to M. floridensis as a “diploid meiotic parthenogen.” As far as I am aware, it is not well established that this species reproduces through meiotic pathenogenesis, and no mention of what evidence is available is presented. Since this mode of reproduction is stressed in the paper, published evidence for this mode of reproduction should be included in the introduction. I am only aware of one reference with any information on Mf reproduction (Handoo et al 2004). This reference reports, on the basis of cytology studies, that a defective first meiotic division occurs. However, they do not see evidence of a second maturation division to form haploid nuclei. The authors need to clarify what is known about the reproduction mechanism of this species. Previous work on cytogenetics (eg, Triantaphyllou, 1985) is over-simplified to the extent that it is misleading. For example, according to that review, there are isolates of M. hapla that reproduce by mitotic parthenogensis, and some isolates of M. incognita appear to be diploid.

In sum, the genome wide comparisons presented here add valuable new insights into how this important group of plant pathogens evolved. These comparisons are well presented with useful figures. The manuscript is quite repetitive and would be easier to read with some reduction. There is oversimplification of what is known about reproduction in RKN. Based on this, the authors have over-interpreted their data in some cases. This can be adjusted in the text to produce an improved manuscript.

Experimental design

One issue concerns the source of the DNA that was used to produce the genome sequence. Since the DNA was extracted from several egg masses and the material is likely not inbred, one cannot be certain whether the polymorphisms in sequence occur within an individual or are segregating in the population. This issue does not negate the work, but should be revealed to readers. Is it known whether reproduction is parthenogenetic or sexual (role of males)? If meiosis occurs, how is heterozygosity maintained?

Nevertheless, I think Mf was a good choice and the evidence presented here supports a role for this species (or a close progenitor) as the origin of M. incognita and likely other hybrid, parthenogenic RKN species.

Validity of the findings

The genome sequence determination and comparative analysis appears to be well done and the comparisons of data from the three species are clearly presented.

Additional comments

First sentence of abstract: Meloidogyne is a genus - all RKN are in this genus. I suggest adjusting the sentence.
Lines 11-12: see comments for abstract. This sentence is awkward and grammar is incorrect.
L2: significant influence on what?
L32: format refs
L43: explain and reference reproductive mode of Mf.
L47: rethink this. Not full meiosis. Also, some isolates of the species M. hapla are meiotic and some are mitotic.
L51. Hybridization with M. fl (perhaps add ‘or a common ancestor’).
L82-83. Unclear: what ‘species is known to reproduce via sexual recombination’? M. enterolobii reproduces by mitotic parthenogenesis (which doesn’t fit your model) and previously it was suggested that M. fl is parthenogenetic.
L87-L110: This section should be adjusted after considering issues above.
L208: to be advised?
L211-229: Much of this page is not results of this work and is repeated in the discussion. I suggest deleting those parts here. Were any studies done for longer range synteny with long contigs?
L256-259: Check sentence structure – I can’t follow.
L391: homologous
L420 and on: This part of the discussion is an essay/review onto itself. However, it is well written and interesting though it does ramble on.

Reviewer 2 ·

Basic reporting

The article is written in clear English, includes sufficient introduction and discussion, and the figures are appropriately labeled and described. Authors understand their research field well, and place their research into the appropriate context.

Experimental design

The paper describes the sequencing of the genome of the root knot nematode Meloidogyne floridensis, in order to establish whether this species is a parental species to the putative hybrid species M. incognita. The research question is made clear, and five alternative evolutionary scenarios are set forward. The data is analyzed, and the discussion evaluates which scenario is best supported by the data.
The investigation has been conducted rigorously, to a high technical standard, and used appropriate software tools and analysis methods. Methods are clearly described with sufficient information to be reproducible by another investigator.

Authors have not fully attempted to conduct statistical hypothesis testing of the hypotheses they set forward, instead choosing to analyze the data to see which “pattern” seems most similar to the expected pattern. An alternative approach – which admittedly is very much more complicated, is to statistically model the different scenarios, generate distributions of data and eliminate scenarios with significantly worse likelihood to explain the observed data. I understand if the authors feel that such phylogeographical modeling is outside the scope of this paper, and it is somewhat unclear how helpful it would be, given the other limitations of their data.

Validity of the findings

Most of the data used is/will be made available to the public. See specific comments below:

165 The final assembly file is available as a blast database
166 and fasta download at www.meloidogyne.org and meloidogyne.nematod.es.
I srongly recommend that the genome data is also deposited in an independent data repository, such as GenBank.
207 Treefiles and scripts for processing trees can be
208 obtained from DataDryad accession [to be advised].
I have not been able to check the data in the Dryad repository, as the accession number is not provided. In addition to tree-files, I’d recommend that the authors deposit fasta-sequences of the sequences they used, along with the alignment of clusters.

Additional comments

The authors have chosen to discuss a controversial topic; that of hybrid speciation in animals. They do a thorough review of the field, but their conclusions may nonetheless be considered controversial by some, as it is challenging textbook knowledge.

I agree with the authors, that given their data (3 genomes of root knot nematodes), and their analysis method; looking at gene and genome duplications, their conclusion that hybrid speciation has occurred in root knot nematodes is correct. There are clearly limitations of the data; the genomes are imperfect, they have only sampled one specimen/genome from each species, and they have to resort to bioinformatic simplifications to distinguish alleles from gene-copies. But the paper is still important as a stepping-stone in understanding the evolution of this important pathogen.

I am somewhat unfamiliar with this particular biological system, but I do think there is room for somewhat more caution than is currently expressed by the authors. For instance, it is not clear to me that M. incognita and M. floridensis are really two different species, from the description provided by authors:
35 Meloidogyne floridensis is a plant pathogenic root knot
36 nematode that was originally characterized as M. incognita, but has since been described as a
37 separate species on the basis of its morphology and a unique esterase isozyme pattern
38 (Jeyaprakash et al. 2006; Handoo et al. 2004).

I’ve read the two mentioned references, and I’m not completely convinced of their conclusion that these are two different species, as there are confounding geographical aspects in their analyses. There are also other “species” of root knot nematodes that can complicate the mix, including M. arenaria, M. chilwoodi and M. javanica. In this case, I have the feeling that the confusing systematics may well be a result of the process the authors describe and/or a few others; hybrid speciation, genome duplication, semi-permeable species boundaries, local host adaptation and non-intuitive phylogeography, because of human displacement of crops. In short, I don’t think the question of which species has hybridized with which will be settled until we’ve sequenced the genomes of a worldwide set of very many root knot nematodes from very many host plant. Such an endeavor is clearly outside the scope of this paper, and the lack of that data doesn’t diminish the value of the research undertaken by the authors. I would however be inclined to be a little bit more cautious in my conclusions than the authors presently are, given our current limited understanding of this biological system.

This is clearly a very interesting system, and a good candidate system for hybrid speciation in animals. I’m looking forward to see future research projects in the field of Meloidogyne evolution, which continue to explore the evolution and phylogeography of these remarkable creatures.

---

## Round 0.2 · Minor Revisions

Dear Dave,

Thanks for revising your article. I think that you have addressed the referees' comments adequately. I won't need to send this revised version back to them, but I'd like some of the clarifications from your rebuttal letter to make their way into the manuscript.

In particular, I think that the concern about intraspecific polymorphism in your sample is sufficiently pertinent to warrant inclusion (and deflection) in your text. Less central but also of relevance is the question about maintenance of heterozygosity; again, your response seems informative to me and could be weaved into the ms.

I hope you agree and are able to implement this quickly. I'll strive to process your revisions without delay!

---

## Round 0.3 · accepted · Accept

Thanks in turn for your swift turnaround time. I am happy to accept this version.